# Study on the relationship between microbial composition within obstructive biliary stents and the severity of obstruction and duration of stent placement

Ichiro Sugawara[1], Yosuke Kawahara[2], Lena Takayasu[3], Kimio Isshi[2], Masayuki Kato[2], Shingo Ono[2], Yuko Hara[2], Toshiki Futakuchi[2], Hiroto Furuhashi[2]*, Rina Kurokawa[3], Kazuki Sumiyama[2], Wataru Suda[3]*

1 Division of Gastroenterology and Hepatology, Department of Internal Medicine, The Jikei University School of Medicine, Tokyo, Japan, 2 Department of Endoscopy, The Jikei University School of Medicine, Tokyo, Japan, 3 Laboratory for Microbiome Sciences, RIKEN Center for Integrative Medical Sciences, Kanagawa, Japan

* wataru.suda@riken.jp (WS); ms04furuhashi@jikei.ac.jp (HF)

**Data Availability Statement:** The data underlying the results presented in the study are available in DDBJ at www.ddbj.nig.ac.jp/biosample/index-e.

## Abstract

Biliary stent occlusion is due, in part, to biofilm formation by bacteria. However, previous culture-based approaches may not have revealed all microorganisms on the surface. Twenty-seven patients underwent endoscopic retrograde biliary drainage for the removal or replacement of plastic biliary stents. We analyzed occlusion severity using image-analyses of a longitudinal section of the biliary stent and evaluated the microbial profile of sludge deposition inside the stents using 16S rRNA sequencing with a MiSeq Illumina platform. We then evaluated the association of microbial profiles with the duration of stent placement and stent occlusion severity. Actinobacteria and Synergistetes were much more abundant in occluded stents compared with non-occluded stents. An abundance of *Bifidobacterium spp.* and OTU00006 *Bifidobacterium animalis* (100%) correlated with stent occlusion severity (rho, 0.62; p<0.001; and 0.42; p = 0.03, respectively), and this relationship remained after adjusting for the duration of stent placement (p = 0.03 and 0.05, respectively). The genus *Bifidobacterium* and *Bifidobacterium animalis* were associated with the degree of occlusion in plastic biliary stents.

## Introduction

The use of a biliary stent can relieve obstructive jaundice [1]. Clinically, a plastic stent will be used in a patient with jaundice as a first treatment to either improve their acute obstructive cholangitis or allow the patient to proceed to the next step, such as surgery and chemotherapy. A plastic stent provides significant benefits by being easy-to-use and as a low-cost endoscopic device.

Presently, there is no effective way to prevent the blockage of the inserted bile duct stent itself, and replacement is a prerequisite. When obstruction occurs, jaundice and cholangitis

html with BioSample accession number
SAMD00262498-SAMD00262524.

**Funding:** This study was supported by the
Japanese Foundation for Research and Promotion
of Endoscopy, Grant 2018. The funders had no role
in study design, data collection and analysis,
decision to publish, or preparation of the
manuscript."

**Competing interests:** The authors have declared
that no competing interests exist.

develop, and the obstructed stent must be removed and replaced with a new one as soon as possible. Because cholangitis is sometimes fatal, it is recommended that plastic stents be replaced with new ones periodically every three months [2–4]. These emergency and routine procedures increase the patient's physical burden and medical costs.

Stent occlusion can be attributable to the deposition of biliary sludge into the plastic stent. The biliary sludge is essentially the aggregate of organic matter such as bilirubin-calcium, cholesterol crystal, palmitic acid, and dietary fiber. Biofilm formation by intestinal bacteria can impact the function of the stent [5]. and bacterial attachment was found to promote biliary sludge formation *in vitro* [6, 7].

Many gut bacteria are hardly ever cultured *in vitro*, which means the bacterial profile found *in vivo* often has negligible representation in conventional culture methods. In recent years, 16S rRNA amplicon sequencing has allowed us to obtain exhaustive bacterial DNA sequences that has revealed an association between intestinal bacteria and particular diseases [8]. However, the relationship between the level of stent occlusion and the bacterial profile was unknown. In addition, it was unclear whether the bacteria in the stent increase or not in dependence on the duration of the placement.

In this study, we used 16S rRNA sequencing to determine the bacterial profile in the stent and to investigate the relationship between microbiota, the occlusion level, and the duration of the stent placement. If the 16S rRNA profile in an occluded stent can be clarified and information obtained to help clarify the causal relationship of occlusion, it is expected to lead to advance prediction of cases of frequent occlusion and the development of stents that are less prone to occlusion.

## Materials and methods

### Study design and patient recruitment

This was an exploratory study using prospectively collected human samples. The study included 27 patients who underwent endoscopic retrograde biliary drainage by plastic stent replacement due to both periodic exchanges (e.g., on a regular 3-month basis) and clinical events (e.g., jaundice or cholangitis) from August 1, 2017 to November 20, 2018. We excluded cases in which information about the stent was unknown because the stent had been inserted at another hospital, cases in which aseptic specimen collection was difficult due to the absence of laboratory technicians, and cases in which consent could not be obtained. As a result, samples from 27 consecutive cases were collected and enrolled in the study. The present study was performed according to the Helsinki Declaration of 1975 and the ethical guidelines for medical and health research involving human subjects (Ministry of Health, Labour and Welfare, Japan). The institutional review board at The Jikei University School of Medicine (No. 29–045 [8661] for collecting human biological samples and 16S rRNA sequencing; and No. 31–435 [10017] for statistical and bioinformatic analysis of clinical and metagenome data) approved the study protocol. This study was registered at University Hospital Medical Information Network Clinical Trials Registry (UMIN-CTR) No.000045806 for enrolling patients and collecting samples, and No.000043829 for microbiome analyses. Statistical and microbiome analysis was performed using cryopreserved samples with a retrospective design after passing ethical review.

### Sample collection, DNA preparation and 16S rRNA amplicon sequencing

Every stent was frozen in nitrogen liquid in a 50-mL Falcon tube immediately after being withdrawn through the duodenoscope channel with forceps, then stored at -80°C. DNA preparation was as follows: 1) the 50-mL Falcon tube housing the extracted stent was thawed on ice; 2)

after opening the Falcon tube on a clean bench, the stent was split into halves longitudinally, then cut every 3 cm cross-sectionally using a sterile scalpel; 3) the longitudinal cutting plane was photographed for image-analysis; and 4) biliary sludge was scraped from the lumen of the biliary stent using a sterile spatula, then transferred into a 2-mL Eppendorf microtube. Then, bacterial DNA was extracted using an enzymatic lysis method, as described previously [9].

The V1-V2 region of 16S rRNA was amplified (20 cycles) by polymerase chain reaction with a universal primer (27Fmod and 338R) and a unique 8-bp barcode, then amplicons were sequenced with a MiSeq Reagent Kit v3 (600 cycles). Reads were assembled for each sample using the barcode sequence. Reads that lacked primer sequences at both ends, had quality values below a threshold of 25, or were suspected to be chimeric sequences due to an alignment complementarity of less than 90% compared with the reference sequences, were eliminated. Following removal of the primer sequence, 3 000 reads were randomly chosen for each sample. The reads were sorted based on quality values and clustered into operational taxonomic units (OTUs) with a threshold of 96% pairwise-identity using the UCLUST algorithm ver.5.2.32. Then, the GLSEARCH program blasted the representative sequence of each OTU to our 16S database (described below) and determined the closest taxa. Our 16S database was composed from three public databases: Ribosomal Database Project (RDP) v.10.31, CORE (http://microbiome.osu.edu/ [last accessed 31 January 2017]), and the sequence database of the NCBI FTP site (ftp://ftp.ncbi.nih.gov/genbank/ [last accessed December 2011]). The OTUs were assigned to the taxa at the phylum and genus level using the identity-threshold of 70% and 94%, respectively.

## Image-analysis of biliary stent occlusion severity

Image-analyses of the sludge formation in occluded stents were evaluated to assess the severity of occlusion. Images were taken every 3 cm along the longitudinal cutting plane of each stent. The "occlusion level (%)" was then evaluated using image-analysis software (ImageJ 1.52q, National Institutes of Health, USA) as follows: 1) the inner lumen of each stent (**Fig 1A**) and its sludge formation (**Fig 1B**) was manually traced using the "polygon selection" or "freehand selection" tool; 2) the area (pixel) for each region of interest was calculated using the "measure"

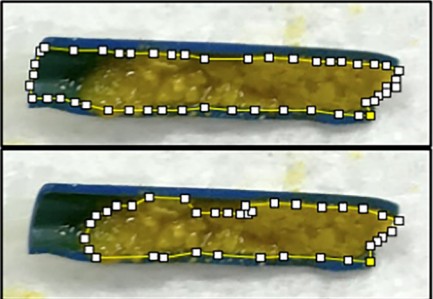
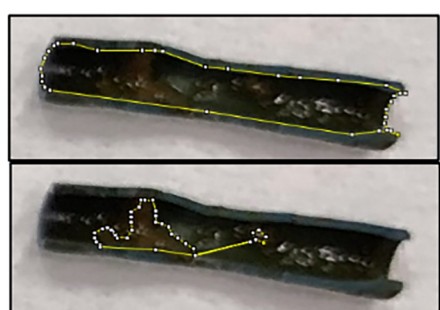

**Fig 1. Photographs of image-analyses of a representative case in the highly and poorly occluded stents.** The picture was taken using the ImageJ 1.52q software (National Institutes of Health, USA) to evaluate the quantity of sludge formation in the occluded stent. The area surrounded by yellow lines with white dots represents the overall inner stent (**a**) and sludge formation (**b**) manually traced using the "polygon selection" or "freehand selection" tool.

tool; and 3) the area of sludge formation was divided by the area of the stent's overall inner lumen, which resulted in the percent occluded area for each lumen along the long axis.

## Outcomes measured

We measured the microbial profile of the biliary sludge formation: 1) α-diversity indices including the number of observed OTUs, Shannon index, abundance-based coverage estimator (ACE), and Chao 1 index for α-diversity; and 2) β-diversity via weighted and unweighted principal coordinate analyses (PCoA) of UniFrac distances. To assess the association between stent occlusion level and clinical/microbial profiles, we measured the correlation among 1) the occlusion level (%); 2) relative abundance (%) of OTUs; and 3) clinical data (duration of stent placement [days], total bilirubin [mg/dl], and C-reactive protein [mg/dl]).

## Statistical analysis

Categorical data were described using frequencies (%) with 95% confidence intervals; continuous data were described using the median value with interquartile ranges for non-normal distribution. For differences between two groups, we used the Fisher's exact test for categorical data, the Mann-Whitney U-test for independent continuous data, and the Wilcoxon signed-rank test for non-independent continuous data. To assess differences in continuous values among three or more groups, we applied the Kruskal-Wallis test for independent data and the Friedman test for non-independent data. Then, if significant, we performed the Wilcoxon signed-rank test for each pair with a false discovery rate adjustment method. To assess α-diversity which summarizes the distribution of abundances in each sample into a single number that depends on evenness and richness, we used the number of OTUs, Shannon index, and Chao 1 index. We used PCoA and permutational multivariate analysis of variance (PERMANOVA) of UniFrac distances for β-diversity which quantifies similarities or dissimilarities between samples. Correlation analyses were performed using Spearman's rho statistic. As a sensitivity analysis, we evaluated the effect of the use of antibiotics and duration of stent placement via sub-group analysis, multi-regression analysis, and matched-pair analysis. Two-tailed tests with a significance level of 5% were used for all tests. For the sample size calculation, assuming that at least one of the top 30 genera in terms of abundance showed Spearman's rho = 0.7, the median of 0.6–0.8, which is defined as a "strong" correlation, we substituted α = 0.05 and power = 0.8, and calculated n = 24.119. Therefore, considering dropouts and sequence failures, 27 cases were included. All statistical analyses were performed using R version 4.0.0 software (R Foundation for Statistical Computing).

## Results

### Background characteristics of patients

A total of 27 patients with 27 biliary stents were enrolled in the study. The underlying conditions of initial biliary obstruction were choledocholithiasis (n = 21), benign strictures (n = 3), gallbladder cancers (n = 2), and cholangiocarcinoma (n = 1) (**Table 1**). The subjects in the present study were predominantly elderly (78 years), male (70%), had gallstones as an underlying disease (79%), and had polyurethane stents inserted (85%), with an average length of stent implantation of 82 days.

### Differences in microbial abundance

**Fig 2A and 2B** shows the individual bacterial abundance (%) at the phylum and the genus level, respectively. The relative abundance of the phylum Actinobacteria and Synergistetes

Table 1. Background characteristics of enrolled patients.

| | | Overall patients (n = 27) |
|---|---|---|
| Age, mean (SD), y | | 77.8 (9.0) |
| Sex, n (%) | Female | 8 (29.6) |
| | Male | 19 (70.4) |
| Use of antibiotics, n (%) | No | 20 (74) |
| | Yes | 7 (26) |
| Use of proton-pump inhibitors, n (%) | No | 21 (78) |
| | Yes | 6 (22) |
| Underlying disease, n (%) | Bile duct stones | 21 (78.6) |
| | Benign stricture | 3 (14.3) |
| | Gallbladder cancer | 2 (7.1) |
| | Cholangiocarcinoma | 1 (0.0) |
| Total bilirubin, mean (SD), mg/dL | | 1.1 (1.5) |
| C-reactive protein, mean (SD), mg/dL | | 2.1 (4.5) |
| Number of previous biliary drainages, mean (SD), n | | 0.9 (1.3) |
| Main materials of plastic stents, n (%) | Polyurethane | 23 (85) |
| | Polyethylene | 4 (14) |
| Duration of stent placement, mean (SD), days | | 82 (85) |

SD, standard deviation

appeared more frequently in the highly occluded stents on the left side. The relative abundance of the genus *Bifidobacterium* (orange bar), *Pyramidobacter* (light green bar), and *Actinomyces* (dark red bar) appeared more frequently in the highly occluded stents.

## Alfa-diversity indices

All the α-diversity indices (number of observed OTU, Chao 1 index, ACE, and Shannon index) tended to be higher in the above than below the 50 percentile of the image analysis-based occlusion level, but no significant difference was shown (**Fig 3**).

## UniFrac PCoA (β-diversity)

**Fig 4** shows the PCoA of UniFrac distances. Both unweighted and weighted UniFrac tended to separate from each group. The PERMANOVA analysis according to UniFrac metrics showed differences between the above (read dots) and below (blue dots) the 50 percentile of the image analysis-based occlusion level (unweighted UniFrac, $R^2 = 0.067$ [p = 0.003]; weighted UniFrac, $R^2 = 0.081$ [p = 0.059]).

## Correlation analysis between microbial profiles of stent sludge and severity of stent occlusion and clinical characteristics

A Spearman's correlation coefficient was calculated between microbial profiles and stent occlusion level (%) calculated by image-analysis of biliary sludge (**Fig 5**; see **S1 Fig** and see **S1 Table**). The genus *Bifidobacterium* and *Pyramidobacter* showed a significant positive correlation (rho = 0.62 [p<0.001], and rho = 0.54 [p = 0.003], respectively). Each of the four diversity indices (Chao 1, observed OTUs, ACE, Shannon index) increased with stent occlusion level (%). In a correlation analysis between microbial profiles and the extended duration of stent placement (see **S2 Table**), extended stent placement was associated with a greater microbial

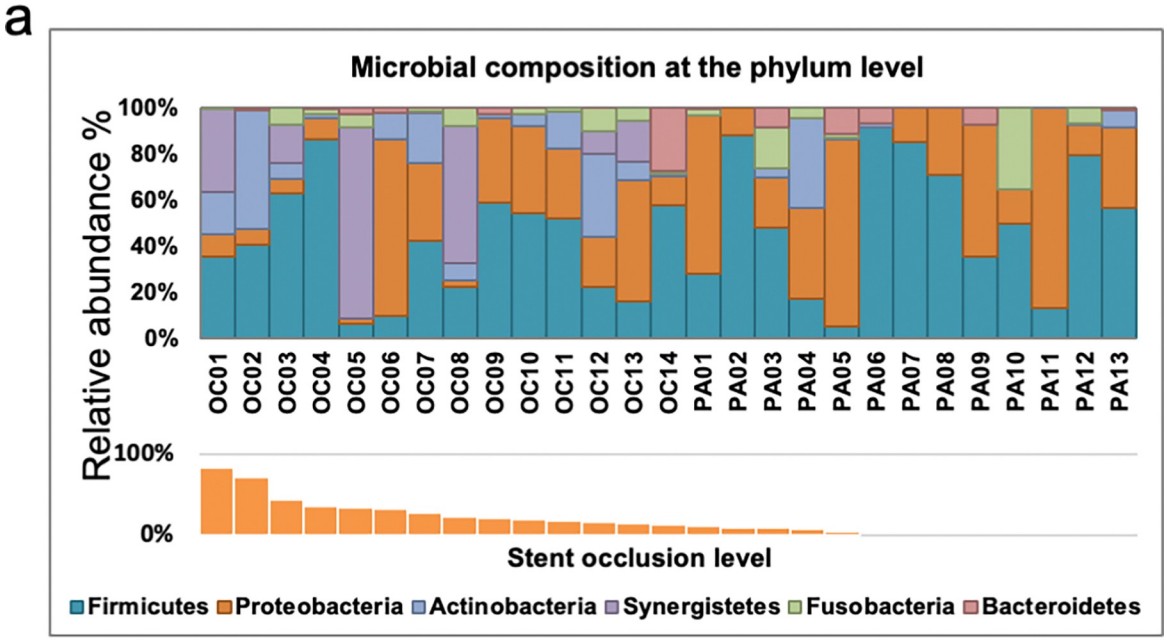

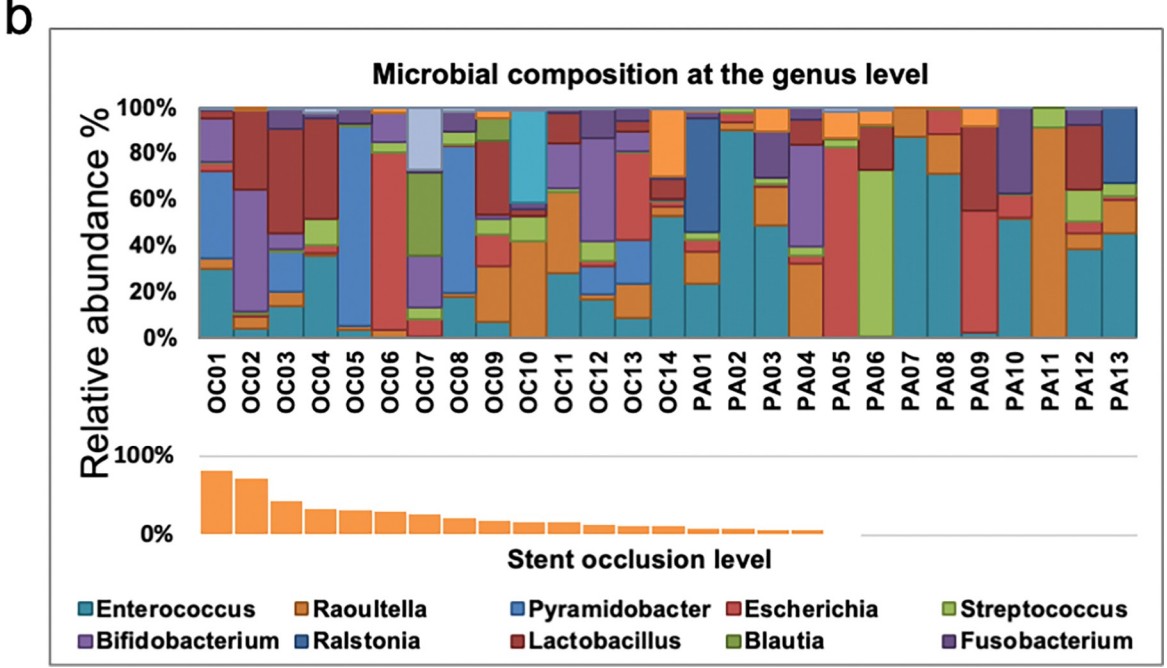

**Fig 2.** The relative abundance of microbial taxonomies at the (a) phylum and (b) genus level. Microbial compositions in each of the 27 individuals are arranged from right to left in increasing order of occlusion severity.

diversity in the biliary stent. The abundance of the genus *Pyramidobacter, Dialister, and Bifidobacterium* increased with the duration of stent placement (rho = 0.57 [p = 0.002], rho = 0.56 [p = 0.003], and rho = 0.51 [p = 0.006], respectively). The results of the above two analyses imply that the duration of stent placement could confound the relationship between microbial profiles and stent occlusion severity. Thus, we performed a multi-regression analysis using the

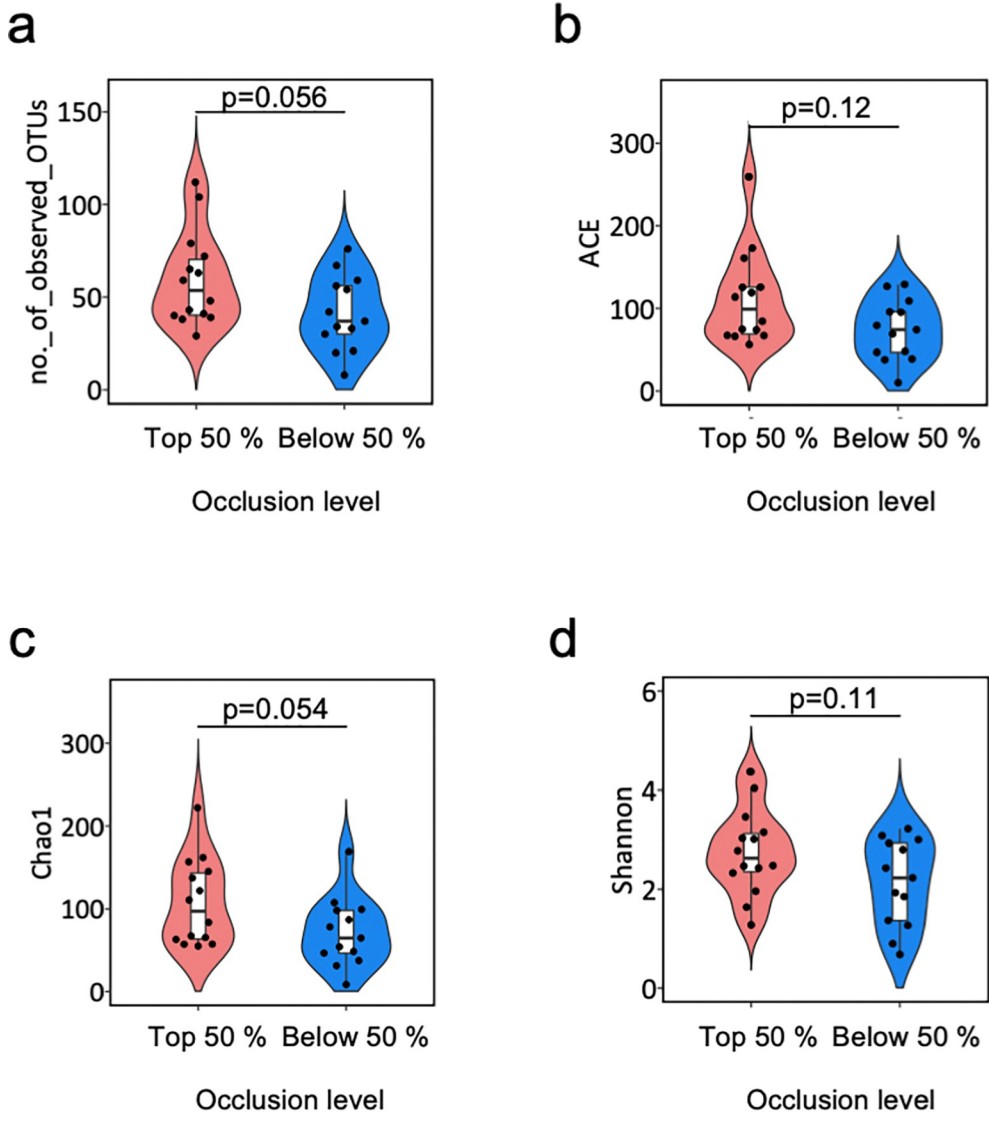

**Fig 3. Comparison of the α-diversity indices between the highly and poorly occluded stents.** Violin plots represent the probable density of the number of observed OTUs (**a**), ACE (**b**), Chao1-based estimated number of OTUs (**c**), and Shannon index (**d**). The boxplots in white represent the 25th and 75th percentiles, the line inside denotes the median, and the whiskers extend to the maximum and minimum values within 1.5 times the difference between the 25th and 75th percentiles. The p-value was calculated by the Wilcoxon rank-sum test. OTU, operational taxonomic unit; *ACE*, abundance-based coverage estimator.

placement duration and the occlusion severity as explanatory variables (see **S3 Table**). The results suggest that Actinobacteria and *Bifidobacterium spp.* were associated with occlusion severity regardless of stent placement duration. In contrast, the relationship with genus *Pyramidobacter* and *Dialister* was placement duration-dependent.

## Sensitivity analyses

The influence of antibiotic use on microbial profiles was analyzed. In a subgroup analysis for cases without antibiotic use (n = 20), most of the correlation coefficients (Spearman's rho) between microbial profiles and the occlusion level did not differ (see **S4 Table**). Meanwhile,

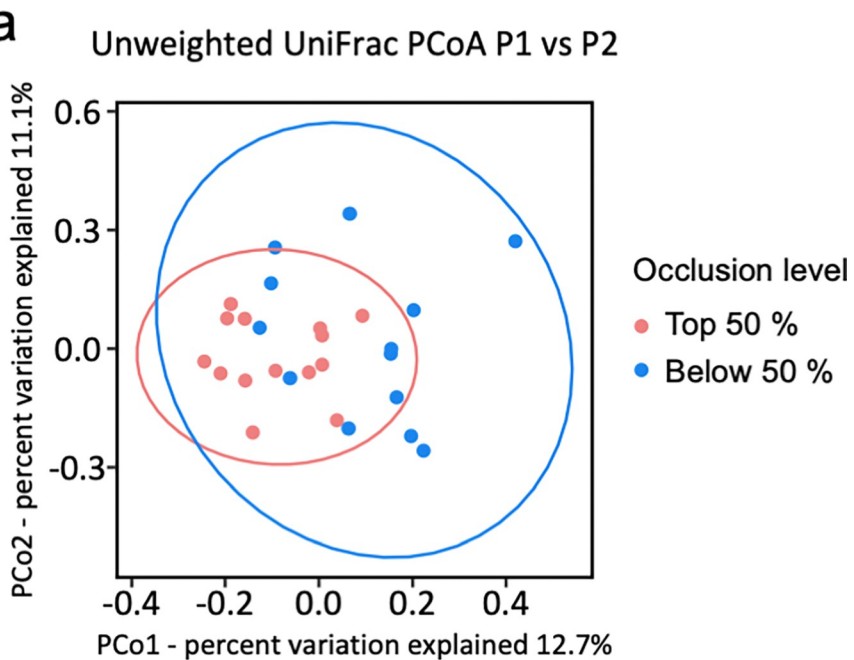

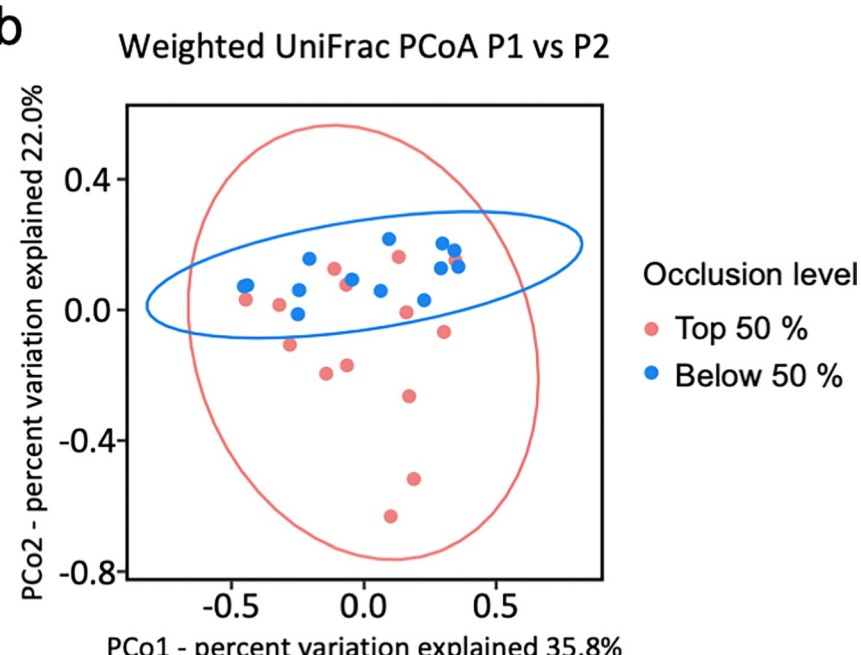

**Fig 4. UniFrac PCoA distance metrics-based analyses (β-diversity).** Weighted (a) and unweighted (b) UniFrac PCoA based on two-dimensional sample plotting using PCo1 and PCo2. The ellipses around the dots represent 95% confidence intervals considering a multivariate normal distribution. The p-value was calculated by PERMANOVA ('*adonis*' function in '*vegan*' R package). *PERMANOVA*, permutational multivariate analysis of variance analysis.

## a. Phylum level

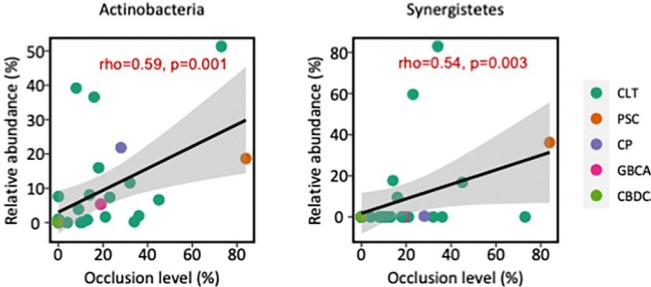

## b. Genus level

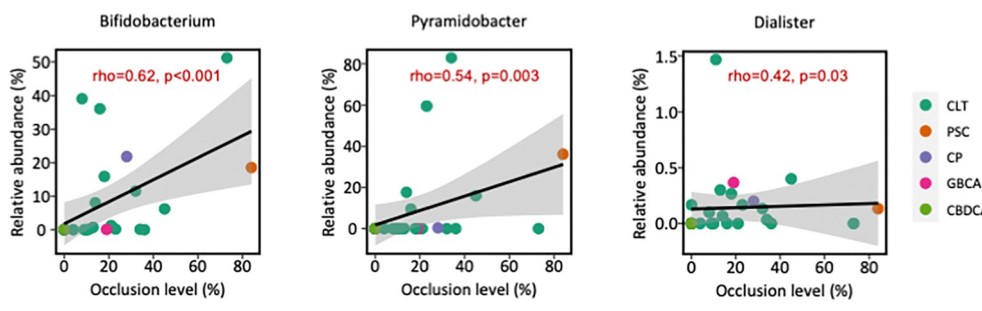

## c. OTU level

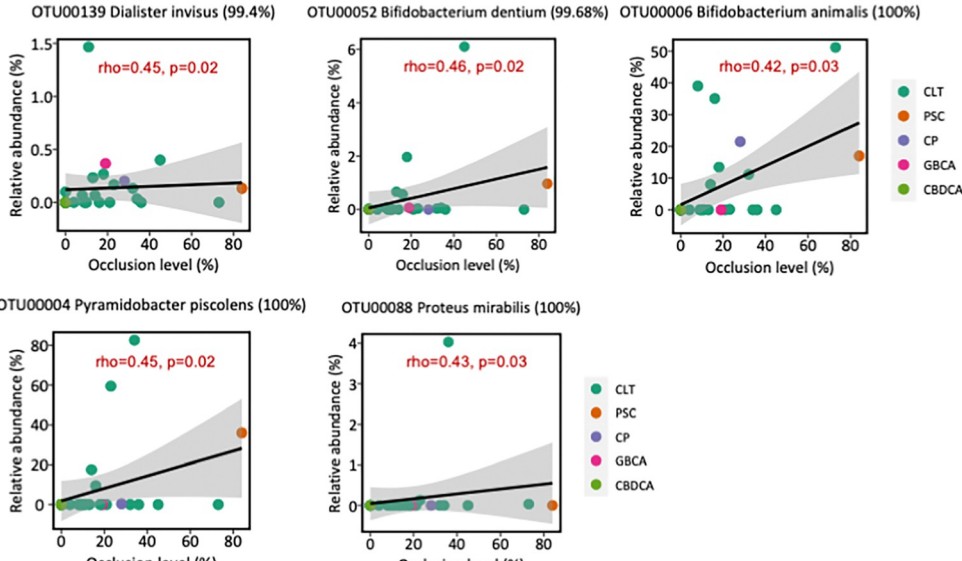

**Fig 5. Correlation between the stent occlusion level (x-axis) and core microbial abundance (y-axis) with statistical significance (p<0.05 [Spearman's test]).** The black line in each scatter plot represents a linear regression line; the region in grey denotes 95% CIs. The color of each dot represents the disease that caused the biliary obstruction at the time of the initial endoscopic drainage. CLT, choledocholithiasis; PSC, primary sclerotic cirrhosis; CP, chronic pancreatitis; GBCA, gall bladder cancer; CBDCA, common bile duct cancer.

stent placement duration was affected by antibiotic use (see **S5 Table**). When limited to patients not treated with antibiotics, there was no trend for *Bifidobacterium spp.* to increase consistent with the duration of stent placement.

## Discussion

This study is the first to attempt to elucidate the relationship between the microbial profile in biliary stents using 16S rRNA amplicon sequencing and the image analysis -based stent occlusion level and duration of stent placement. We found biliary sludge inside the stents harbored a broad range of anaerobic bacteria with high diversity, which was notably different from the results of previous culture-dependent studies. The correlation analysis demonstrated that prolonged stent placement and increased sludge formation significantly correlated with a higher abundance of certain anaerobic bacteria such as *Bifidobacterium*, *Pyramidobacter* and *Dialister*.

These results are strikingly different from preceding culture-dependent studies, where only a few major aerobes, such as *Escherichia coli*, *Klebsiella pneumoniae* and *Klebsiella oxytoca*, were isolated and related to biliary stent clogging [10, 11]. The present study's findings, however, suggest the sludge harbors a wider variety of microbes of approximately 50 OTUs, and that occlusion severity is associated with several anaerobes that were not reported in prior studies.

This difference may be attributable to the difference in analytic methods for microbes. Compared to culture-based methods, 16S rRNA amplicon sequencing can comprehensively identify bacterial profiles [12–14]. The recent refinement of nucleotide sequencing databases of 16S ribosome regions has further improved accuracy [14–17]. This highlights the limitations of culture-based methods where they can fail to culture obligate anaerobic bacteria [18].

Previous studies focusing on dental caries and periodontal diseases revealed that *Bifidobacterium* spp. plays an important role in biofilm formation in dental plaque via acidogenicity and aciduricity [19, 20]. Our data showed that *B. animalis*, *B. dentium* and *B. longum* were mainly observed in OTU-level analyses under the genus *Bifidobacterium*. According to previous reports, these microbes harbor superior acidogenic potential compared with other bacteria and promote a higher pH drop in the environment in the presence of glucose, lactose and other oligosaccharides [21–23]. Notably, an *in vitro* experiment demonstrated that strains of *B. animalis* boosted the ability of biofilm formation under bile conditions [19]. Meanwhile, *Pyramidobacter piscolens*, another main OTU that was associated with occlusion level in this study, also has acidogenicity where acetic acid, isovaleric acid and hydrogen sulphate are produced as a result of metabolism. Many bacteria identified in our analysis have acid production ability, implying the possibility that acidogenicity is a major contributor to biofilm formation in the basic environment inside the stent.

It is interesting to note that, except for *Proteus mirabilis*, the predominant anaerobes in the occluded stents are considered commensal bacterium in the oral cavity or upper digestive tracts. A 16S rRNA sequencing-based study demonstrated that microbial composition in the upper digestive tract resembles the salivary microbiota [24]. Among the taxa associated with occlusion severity, the genera *Pyramidobacter* and *Dialister*, and *Bifidobacterium dentium* are known as commensal oral bacteria. These results imply that the biliary sludge microbiota may have originated in the oral cavity via flow back from the duodenum.

This study, however, does have several limitations. Firstly, there is a chance of contamination at the time of withdrawing the biliary stent. All experimental procedures were performed under sterile conditions except for the stent removal. Despite careful maneuvering to minimize contaminants, the endoscopic withdrawal of the biliary stent through the upper digestive

tract was unavoidable, possibly resulting in fluid contamination in the upper digestive tract. However, ambient bacteria would be less likely to contaminate bacteria inside the biliary stent due to it being an enclosed environment. The probability of contamination would be similar between severe and mild occlusion so the detection of more abundant microbes in the occluded stent is unlikely to be the result of contamination. On the other hand, the genus *Ralstonia* was the only bacterium inversely correlated with obstruction. In the study, several non-occluded stents showed extremely small amounts of biliary sludge, in which case the effects of contamination could be more pronounced. Some reports have demonstrated that *Ralstonia* is a constituent contaminant genus in the process of ribosome sequencing [25, 26]. A second potential limitation is that more patients with the non-occluded stents received antibiotic treatment within 28 days, which might lead to a difference in microbial composition. A sensitivity analysis, however, which examined the impact of antibiotics suggested that antibiotic use did not significantly affect the more dominant bacterial composition in the occluded stent. Thirdly, the results of this study do not provide evidence for a direct causal relationship between the identified microorganisms and biliary stent obstruction. The relationship between the organisms responsible for cholangitis and sludge formation remains unknown. In the present study, blood cultures were performed in 5 of the 27 cases, 3 were negative, and *Escherichia coli* was detected in the remaining 2 cases. Remarkably, in these 2 cases, *E. coli* was not necessarily the top species according to the 16S sequencing for the stent sludge. Based on the above, the causative organism of cholangitis might be different from the causative organism of sludge formation. Further trials are needed to detect biofilm or sludge precipitation on the stent surface using bacterial culture solutions *in vitro* or *in vivo*.

## Conclusion

We have successfully identified the up-until-now unknown microbial composition within occluded biliary stents using 16S rRNA sequencing, where specific anaerobes were significantly dominant with the severity of stent occlusion. Our data raise the important possibility that isolating the identified bacteria may clarify the causal relationship and thereby assist in the selection of the most appropriate prophylactic antibiotics or the development of a drug-coated stent, which may improve prognosis and quality of life, as well as cut costs in medical resources.

## Supporting information

**S1 Checklist. STROBE statement—Checklist of items that should be included in reports of observational studies.**
(DOCX)

**S1 Fig. Correlation between the stent occlusion level (x-axis) and core microbial abundance (y-axis) with statistical significance (p<0.05 [Spearman's test]).** The black line in each scatter plot represents a linear regression line; the region in grey denotes 95% CIs. The color of each dot represents the disease that caused the biliary obstruction at the time of the initial endoscopic drainage.
(PDF)

**S1 Table. The correlation between stent occlusion level (%) and the microbial profile.**
*Spearman's correlation coefficient between each profile and severity of stent occlusion. Occlusion level calculated by image-analysis of biliary sludge inside the stent with Image J platform. OTU, operational taxonomic unit; ACE, abundance-based coverage estimator.
(PDF)

**S2 Table. The correlation between the duration of the stent placement (days) and the microbial profile.** *Spearman's correlation coefficient between each profile and the duration of the stent placement. OTU, operational taxonomic unit; ACE, abundance-based coverage estimator.
(PDF)

**S3 Table. Correlation between stent occlusion level (%) and microbial profile, adjusted for the duration of stent placement using multi-regression analysis.** OTU, operational taxonomic unit; ACE, abundance-based coverage estimator. CI, confidence interval; SE, standard error.
(PDF)

**S4 Table. Differences between stent occlusion severity (%) and microbial profile according to the use of antibiotics (overall cases vs non-antibiotic cases).** OTU, operational taxonomic unit; ACE, abundance-based coverage estimator.
(PDF)

**S5 Table. Differences between duration of stent placement (days) and microbial profile according to the use of antibiotics (overall cases vs non-antibiotic cases).** OTU, operational taxonomic unit; ACE, abundance-based coverage estimator.
(PDF)

## Acknowledgments

We thank C. Shindo, K. Komiya, E. Iioka, and M. Kiuchi (RIKEN) for technical assistance.

## Author Contributions

**Conceptualization:** Yosuke Kawahara, Kimio Isshi, Masayuki Kato, Shingo Ono, Toshiki Futakuchi, Hiroto Furuhashi, Kazuki Sumiyama.

**Data curation:** Lena Takayasu, Shingo Ono, Hiroto Furuhashi.

**Formal analysis:** Lena Takayasu, Hiroto Furuhashi.

**Funding acquisition:** Masayuki Kato, Rina Kurokawa.

**Methodology:** Lena Takayasu, Shingo Ono, Yuko Hara, Hiroto Furuhashi, Wataru Suda.

**Project administration:** Ichiro Sugawara, Kimio Isshi, Toshiki Futakuchi, Hiroto Furuhashi, Rina Kurokawa, Kazuki Sumiyama.

**Resources:** Ichiro Sugawara, Kimio Isshi, Rina Kurokawa.

**Software:** Yosuke Kawahara, Lena Takayasu, Rina Kurokawa, Wataru Suda.

**Supervision:** Kimio Isshi, Masayuki Kato, Yuko Hara, Kazuki Sumiyama, Wataru Suda.

**Validation:** Toshiki Futakuchi, Hiroto Furuhashi, Rina Kurokawa, Wataru Suda.

**Visualization:** Kimio Isshi, Toshiki Futakuchi, Hiroto Furuhashi, Rina Kurokawa, Wataru Suda.

**Writing – original draft:** Ichiro Sugawara, Masayuki Kato, Shingo Ono, Hiroto Furuhashi.

**Writing – review & editing:** Masayuki Kato, Yuko Hara, Toshiki Futakuchi, Rina Kurokawa, Kazuki Sumiyama, Wataru Suda.

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
