## [Decision Letter · Decision Letter 0]

6 Nov 2024

PONE-D-24-29177Microbial composition in obstructed biliary stent is altered depending on the occlusion severity and the duration of stent placementPLOS ONE

Dear Dr. Furuhashi,

Thank you for submitting your manuscript to PLOS ONE. After careful consideration, we feel that it has merit but does not fully meet PLOS ONE’s publication criteria as it currently stands. Therefore, we invite you to submit a revised version of the manuscript that addresses the points raised during the review process.

We look forward to receiving your revised manuscript.

Kind regards,

Wenguo Cui, Ph.D

Academic Editor

PLOS ONE

“This study was supported by the Japanese Foundation for Research and Promotion of Endoscopy, Grant 2018.”

Reviewers' comments:

Reviewer's Responses to Questions

**Comments to the Author**

1. Is the manuscript technically sound, and do the data support the conclusions?

Reviewer #1: Yes

Reviewer #2: Partly

2. Has the statistical analysis been performed appropriately and rigorously? 

Reviewer #1: Yes

Reviewer #2: Yes

3. Have the authors made all data underlying the findings in their manuscript fully available?

Reviewer #1: Yes

Reviewer #2: Yes

4. Is the manuscript presented in an intelligible fashion and written in standard English?

Reviewer #1: Yes

Reviewer #2: Yes

5. Review Comments to the Author

Reviewer #1: This article discusses the correlation between the composition of microorganisms in biliary stent obstruction and the degree of obstruction and the time of stent placement. It is valuable to find the correlation between the species of microorganisms in biliary stent obstruction and the time and degree of obstruction. 16 s rRNA amplification son sequencing can effectively make up for the defect of traditional culture, especially for training and testing of anaerobic bacteria. At the same time, the appropriate and accurate statistical analysis of this study, prospective clinical trial standard, strict, can come to the conclusion. In addition, the authors provide sufficient research data to support the conclusions of the manuscript. Therefore, we suggest that receive the manuscripts. However, several concerns need to be considered by the authors. First, a total of 27 patients were enrolled in this study, which was a small number of patient cases. Suggestion to increase the number of cases, antibiotics, patients with underlying disease in analysis. For example: to investigate the antibiotic use, time whether can affect the kinds of microbial composition in obstructed biliary stent? When bacteria become sensitive to antibiotics, they are eliminated, while bacteria that are resistant or tolerant to antibiotics proliferate and become the dominant strain of infection. Second, this study included patient information about 6 years ago (August 1, 2017 to November 20, 2018). Please make it clear that this is a prospective study rather than a retrospective study. Third, the interval of case information is too long. Are there cases missed, errors, and research selection bias?

Reviewer #2: The authors selected 27 patients and studied the relationship between the microbial composition inside obstructive biliary stents, the severity of obstruction, and the duration of stent placement. The manuscript still has the following shortcomings, and it is recommended to further improve the quality of the manuscript:

1，I believe the title needs to be revised to better suit the requirements of a research paper title, such as: "Study on the Relationship between Microbial Composition within Obstructive Biliary Stents and the Severity of Obstruction and Duration of Stent Placement."

2，It is recommended to further supplement the introduction with information on the causes of biliary obstruction, current treatment methods, and the challenges associated with them.

3, It is advisable to conclude the introduction by highlighting the significance of this study, such as providing guidance from a microbial perspective for early prevention of biliary stent obstruction.

4, It is suggested to further increase the sample size.

5, The manuscript lacks a conclusion section.

6. PLOS authors have the option to publish the peer review history of their article (what does this mean?). If published, this will include your full peer review and any attached files.

Reviewer #1: No

Reviewer #2: No

---

## [Author Response · Author response to Decision Letter 0]

23 Nov 2024

Thank you for taking the time to respectfully review our manuscript. Many of the points the reviewers raised were all important and we believe that by addressing them we have further improved the quality of our manuscript. We also thank you for pointing out the descriptions that do not meet the submission guidelines as journal requirements. We have corrected all of them and hopefully you will take another look at the manuscript.

Reviewer #1: 

Comment no.1:

First, a total of 27 patients were enrolled in this study, which was a small number of patient cases. Suggestion to increase the number of cases, antibiotics, patients with underlying disease in analysis. For example: to investigate the antibiotic use, time whether can affect the kinds of microbial composition in obstructed biliary stent? When bacteria become sensitive to antibiotics, they are eliminated, while bacteria that are resistant or tolerant to antibiotics proliferate and become the dominant strain of infection. 

Our response:

We deeply understand and agree with your concerns about the impact of antimicrobials and other clinical variables on the results. We have in fact been considering the possibility of addressing these effects from the time of the experimental design. However, there are dozens of factors such as antimicrobials, background disease, gender, age, length of indwelling, effect of anti-acids and bowel regulators, presence of other medications, type of stent, presence of clinical infection symptoms, presence of jaundice, and so on. Correcting for all of these would require more than 10 cases per factor, i.e., more than several hundred cases. With few microbiome studies on bile duct stent occluded contents, we decided that it would be difficult to obtain patient consent and funding for more than several hundred cases as an exploratory study, given the uncertain feasibility of such a study. We hope that the results of this study will be understood as the results of an exploratory and preliminary design study that has proven its feasibility. Based on the results of this study, we have obtained results on the amount of DNA that can be collected and the number of reads, and we would like to continue the experiment with more cases in future studies to address these biases. We have revised the manuscript by adding statements in the second limitation, in the Discussion section.

Comment no.2:

Second, this study included patient information about 6 years ago (August 1, 2017 to November 20, 2018). Please make it clear that this is a prospective study rather than a retrospective study. 

Our response:

We apologize for the ambiguity of the description. Samples were prospectively collected with written consent through a biobank study in which endoscopically available specimens were comprehensively allowed to be used for bacterial experiments. However, statistical and bacterial analysis of the microbiome in relation to occlusions in bile duct stents, focusing on the level of occlusion was performed after the sample collection. In other words, strictly speaking, it is appropriate to consider this study a retrospective study. We have revised the manuscript accordingly.

Comment no.3:

Third, the interval of case information is too long. Are there cases missed, errors, and research selection bias?

Our response:

We appreciate your suggestion. Patient recruitment for this study was conducted at only one facility (Jikei University Katsushika Medical Center). The annual number of biliary endoscopies at this facility is about 270. Of these, one-third of the total involved removal of a stent. Of these, we excluded cases in which information about the stent was unknown because a bile duct stent had been inserted at another hospital, cases in which consent could not be obtained, and cases in which aseptic specimen collection was difficult due to the absence of laboratory technicians. In particular, since endoscopic stent removal was often performed at night as an emergency procedure, the absence of lab technicians caused a decrease in the number of cases. We have maintained case continuity for cases meeting the eligibility criterion. We have revised the manuscript in this regard.

Reviewer #2: 

The authors selected 27 patients and studied the relationship between the microbial composition inside obstructive biliary stents, the severity of obstruction, and the duration of stent placement. The manuscript still has the following shortcomings, and it is recommended to further improve the quality of the manuscript:

 1，I believe the title needs to be revised to better suit the requirements of a research paper title, such as: "Study on the Relationship between Microbial Composition within Obstructive Biliary Stents and the Severity of Obstruction and Duration of Stent Placement."

Our response:

Thank you for your meaningful remarks. We have revised the title as you suggested.

 2，It is recommended to further supplement the introduction with information on the causes of biliary obstruction, current treatment methods, and the challenges associated with them.

Our response:

We appreciate your very important suggestions. We have added to the Introduction section a description of the current challenges and coping strategies related to obstructive jaundice and cholangitis practice. We hope that these additions will help readers unfamiliar with cholangitis treatment to understand this article.

3, It is advisable to conclude the introduction by highlighting the significance of this study, such as providing guidance from a microbial perspective for early prevention of biliary stent obstruction.

Our response:

We appreciate very much for the reviewer’s very useful remarks. We have added descriptins at the end of the introduction about the effects of the study results. 

 4, It is suggested to further increase the sample size.

Our response:

We agree with your comments on the desirability of increasing the sample size. We would appreciate it if you could refer to our comments on reviewer number 1's first point.

 5, The manuscript lacks a conclusion section.

Our response:

We apologize for the missing discussion section in the manuscript. We have revised the title as you suggested.

---

## [Decision Letter · Decision Letter 1]

26 Dec 2024

Study on the relationship between microbial composition within obstructive biliary stents and the severity of obstruction and duration of stent placement.

PONE-D-24-29177R1

Dear Dr. Furuhashi,

We’re pleased to inform you that your manuscript has been judged scientifically suitable for publication and will be formally accepted for publication once it meets all outstanding technical requirements.

Kind regards,

Wenguo Cui, Ph.D

Academic Editor

PLOS ONE

Additional Editor Comments (optional):

Reviewers' comments:

Reviewer's Responses to Questions

**Comments to the Author**

1. If the authors have adequately addressed your comments raised in a previous round of review and you feel that this manuscript is now acceptable for publication, you may indicate that here to bypass the “Comments to the Author” section, enter your conflict of interest statement in the “Confidential to Editor” section, and submit your "Accept" recommendation.

Reviewer #1: All comments have been addressed

Reviewer #2: All comments have been addressed

2. Is the manuscript technically sound, and do the data support the conclusions?

Reviewer #1: Yes

Reviewer #2: Yes

3. Has the statistical analysis been performed appropriately and rigorously? 

Reviewer #1: Yes

Reviewer #2: Yes

4. Have the authors made all data underlying the findings in their manuscript fully available?

Reviewer #1: Yes

Reviewer #2: Yes

5. Is the manuscript presented in an intelligible fashion and written in standard English?

Reviewer #1: Yes

Reviewer #2: Yes

6. Review Comments to the Author

Reviewer #1: After revision, the quality of the manuscript met the requirements and was recommended for acceptance.

Reviewer #2: Good modification for publication. The authors have well addressed my concerns. I recommend the acceptance of the manuscript.

7. PLOS authors have the option to publish the peer review history of their article (what does this mean?). If published, this will include your full peer review and any attached files.

Reviewer #1: No

Reviewer #2: No

---

## [Editor Report · Acceptance letter]

30 Dec 2024

PONE-D-24-29177R1 

PLOS ONE

Dear Dr. Furuhashi, 

I'm pleased to inform you that your manuscript has been deemed suitable for publication in PLOS ONE. Congratulations! Your manuscript is now being handed over to our production team.

Kind regards, 

on behalf of

Professor Wenguo Cui 

Academic Editor

PLOS ONE